# RenderGAN: Generating Realistic Labeled Data

**Leon Sixt, Benjamin Wild, & Tim Landgraf**
Fachbereich Mathematik und Informatik
Freie Universität Berlin
Berlin, Germany
`{leon.sixt, benjamin.wild, tim.landgraf}@fu-berlin.de`

## Abstract

Deep Convolutional Neuronal Networks (DCNNs) are showing remarkable performance on many computer vision tasks. Due to their large parameter space, they require many labeled samples when trained in a supervised setting. The costs of annotating data manually can render the use of DCNNs infeasible. We present a novel framework called RenderGAN that can generate large amounts of realistic, labeled images by combining a 3D model and the Generative Adversarial Network framework. In our approach, image augmentations (e.g. lighting, background, and detail) are learned from unlabeled data such that the generated images are strikingly realistic while preserving the labels known from the 3D model. We apply the RenderGAN framework to generate images of barcode-like markers that are attached to honeybees. Training a DCNN on data generated by the RenderGAN yields considerably better performance than training it on various baselines.

## 1 Introduction

When an image is taken from a real world scene, many factors determine the final appearance: background, lighting, object shape, position and orientation of the object, the noise of the camera sensor, and more. In computer vision, high-level information such as class, shape, or pose is reconstructed from raw image data. Most real-world applications require the reconstruction to be invariant to noise, background, and lighting changes.

In recent years, deep convolutional neural networks (DCNNs) advanced to the state of the art in many computer vision tasks (Krizhevsky et al., 2012; He et al., 2015; Razavian et al., 2014). More training data usually increases the performance of DCNNs. While image data is mostly abundant, labels for supervised training must often be created manually – a time-consuming and tedious activity. For complex annotations such as human joint angles, camera viewpoint or image segmentation, the costs of labeling can be prohibitive.

In this paper, we propose a method to drastically reduce the costs of labeling such that we can train a model to predict even complex sets of labels. We present a generative model that can sample from the joint distribution of labels and data. The training procedure of our model does not require any manual labeling. We show that the generated data is of high quality and can be used to train a model in a supervised setting, i.e. a model that maps from real samples to labels, without using any manually labeled samples.

We propose two modifications to the recently introduced GAN framework (Goodfellow et al., 2014). First, a simple 3D model is embedded into the generator network to produce samples from corresponding input labels. Second, the generator learns to add missing image characteristics to the model output using a number of parameterized augmentation functions. In the adversarial training we leverage large amounts of unlabeled image data to learn the particular form of blur, lighting, background and image detail. By constraining the augmentation functions we ensure that the resulting image still represents the given set of labels. The resulting images are hard to distinguish from real samples and can be used to train a DCNN to predict the labels from real input data.

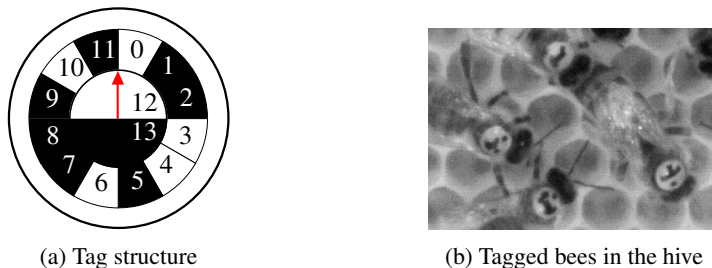

(a) Tag structure (b) Tagged bees in the hive

Figure 1: **(a)** The tag represents a unique binary code (cell 0 to 11) and encodes the orientation with the semicircles 12 and 13. The red arrow points towards the head of the bee. This tag encodes the id `100110100010`. **(b)** Cutout from a high-resolution image.

The RenderGAN framework was developed to solve the scarcity of labeled data in the BeesBook project (Wario et al., 2015) in which we analyze the social behavior of honeybees. A barcode-like marker is attached to the honeybees' backs for identification (see Fig. 1). Annotating this data is tedious, and therefore only a limited amount of labeled data exists. A 3D model (see the upper row of Fig. 2) generates a simple image of the tag based on position, orientation, and bit configuration. The RenderGAN then learns from unlabeled data to add lighting, background, and image details.

Training a DCNN on data generated by the RenderGAN yields considerably better performance compared to various baselines. We furthermore include a previously used computer vision pipeline in the evaluation. The networks' detections are used as feature to track the honeybees over time. When we use detections from the DCNN instead of the computer vision pipeline, the accuracy of assigning the true id increases from 55% to 96%.

Our contributions are as follows. We present an extension of the GAN framework that allows to sample from the joint distribution of data and labels. The generated samples are nearly indistinguishable from real data for a human observer and can be used to train a DCNN end-to-end to classify real samples. In a real-world use case, our approach significantly outperforms several baselines. Our approach requires no manual labeling. The simple 3D model is the only form of supervision.

## 2 RELATED WORK

There exists multiple approaches to reduce the costs associated with labeling.

A common approach to deal with limited amount of labels is data augmentation (Goodfellow et al., 2016, Chapter 7.4). Translation, noise, and other deformations can often be applied without changing the labels, thereby effectively increasing the number of training samples and reducing overfitting.

DCNNs learn a hierarchy of features – many of which are applicable to related domains (Yosinski et al., 2014). Therefore, a common technique is to pre-train a model on a larger dataset such as ImageNet (Deng et al., 2009) and then fine-tune its parameters to the task at hand (Girshick et al., 2014; Long et al., 2015; Razavian et al., 2014). This technique only works in cases where a large enough related dataset exists. Furthermore, labeling enough data for fine-tuning might still be costly.

If a basic model of the data exists (for example, a 3D model of the human body), it can be used to generate labeled data. Peng et al. (2015) generated training data for a DCNN with 3D-CAD models.

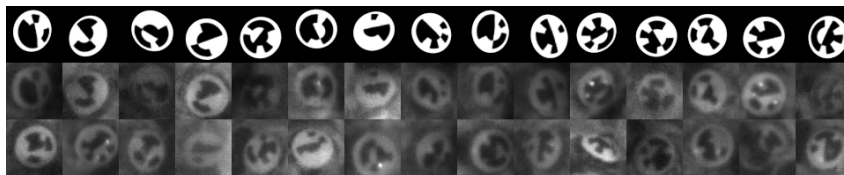

Figure 2: **First row:** Images from the 3D model without augmentation. **Below:** Corresponding images from the RenderGAN. **Last row:** Real images of bee's tags.

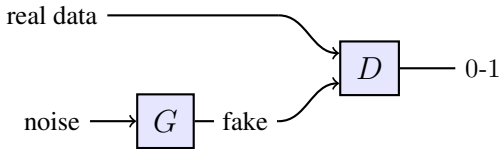

Figure 3: Topology of a GAN. The discriminator network $D$ is trained to distinguish between "fake" and real data. The generator network $G$ receives a random vector as input. $G$ is optimized to maximize the chance of the discriminator making a mistake.

Su et al. (2015) used 3D-CAD models from large online repositories to generate large amounts of training images for the viewpoint estimation task on the PASCAL 3D+ dataset (Xiang et al., 2014). Massa et al. (2015) are matching natural images to 3D-CAD models with features extracted from a DCNN. Richter et al. (2016) and Ros et al. (2016) used 3D game engines to collect labeled data for image segmentation. However, the explicit modeling of the image acquisition physics (scene lighting, reflections, lense distortions, sensor noise, etc.) is cumbersome and might still not be able to fully reproduce the particularities of the imaging process such as unstructured background or object specific noise. Training a DCNN on generated data that misses certain features will result in overfitting and poor performance on the real data.

Generative Adversarial Networks (GAN) (see Fig. 3) can learn to generate high-quality samples (Goodfellow et al., 2014), i.e. sample from the data distribution $p(x)$. Denton et al. (2015) synthesized images with a GAN on the CIFAR dataset (Krizhevsky, 2009), which were hard for humans to distinguish from real images. While a GAN implicitly learns a meaningful latent embedding of the data (Radford et al., 2015), there is no simple relationship between the latent dimensions and the labels of interest. Therefore, high-level information can't be inferred from generated samples. cGANs are an extension of GANs to sample from a conditional distribution given some labels, i.e. $p(x|l)$. However, training cGANs requires a labeled dataset. Springenberg (2015) showed that GANs can be used in a semi-supervised setting but restricted their analysis to categorical labels. Wang & Gupta (2016) trained two separate GANs, one to model the object normals and another one for the texture conditioned on the normals. As they rely on conditional GANs, they need large amounts of labeled data. Chen et al. (2016) used an information theoretic to disentangle the representation. They decomposed the representation into a structured and unstructured part. And successfully related on a qualitative level the structured part to high-level concepts such as camera viewpoint or hair style. However, explicitly controlling the relationship between the latent space and generated samples *without using labeled data* is an open problem, i.e. sampling from $p(x, l)$ without requiring labels for training.

## 3 RENDERGAN

Most supervised learning tasks can be modeled as a regression problem, i.e. approximating a function $\hat{f} : \mathbb{R}^n \mapsto L$ that maps from data space $\mathbb{R}$ to label space $L$. We consider $\hat{f}$ to be the best available function on this particular task. Analogous to ground truth data, one could call $\hat{f}$ the ground truth function.

In the RenderGAN framework, we aim to solve the inverse problem to this regression task: generate data given the labels. This is achieved by embedding a simple 3D model into the generator of a GAN. The samples generated by the simple model must correspond to the given labels but may lack many factors of the real data such as background or lightning. Through a cascade of augmentation functions, the generator can adapt the images from the 3D model to match the real data.

We formalize image augmentation as a function $\phi(x, d)$, which modifies the image $x$ based on the augmentation parameter $d$ (a tensor of any rank). The augmentation must preserve the labels of the image $x$. Therefore, it must hold for all images $x$ and all augmentations parameters $d$:

$$\hat{f}\left(\phi(x, d)\right) = \hat{f}(x) \tag{1}$$

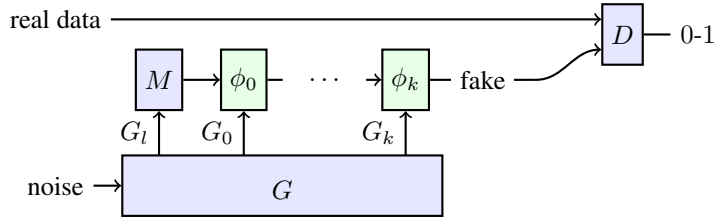

Figure 4: The generator $G$ cannot directly produce samples. Instead, $G$ has to predict parameters $G_l$ for the 3D model $M$. The image generated by $M$ is then modified through the augmentation functions $\phi_i$ parameterized by $G_i$ to match the real data.

The augmentation function must furthermore be differentiable w.r.t. $x$ and $d$ as the gradient will be back-propagated through $\phi$ to the generator. Image augmentations such as lighting, surrounding, and noise do preserve the labels and fit this definition. We will provide appropriate definitions of $\phi$ for the mentioned augmentations in the following section.

If appropriate augmentation functions are found that can model the missing factors and are differentiable, we can use the GAN framework to find parameters that result in realistic output images. Multiple augmentation functions can be combined to perform a more complex augmentation. Here, we will consider multiple augmentation functions applied sequentially, i.e. we have $k$ augmentation functions $\phi_i$ and $k$ corresponding outputs $G_i$ from the generator. The output of the previous augmentation function is the input to the next one. Thus, we can write the generator given some labels l as:

$$g(z, l) = \phi_k(\phi_{k-1}(\ldots \phi_0(M(l), G_0(z)) \ldots, G_{k-1}(z)), G_k(z)) \tag{2}$$

where M is the 3D model. We can furthermore learn the label distribution with the generator. As the discriminator loss must be backpropagated through the 3D model M, it must be differentiable. This can be achieved by emulating the 3D model with a neural network (Dosovitskiy et al., 2015). The resulting generator g(z) can be written as (see Fig. 4 for a visual interpretation):

$$g(z) = \phi_k(\phi_{k-1}(\ldots \phi_0(M(G_l(z)), G_0(z)) \ldots, G_{k-1}(z)), G_k(z)) \tag{3}$$

As any differentiable function approximator can be employed in the GAN framework, the theoretical properties still hold. The training is carried out as in the conventional GAN framework. In a real application, the augmentation functions might restrict the generator from converging to the data distribution.

If the training converges, we can collect generated realistic data with $g(z)$ and the high-level information captured in the 3D model with $G_l(z)$. We can now train a supervised learning algorithm on the labeled generated data $(G_l(z), g(z))$ and solve the regression task of approximating $\hat{f}$ without depending on manual labels.

## 4 APPLICATION TO THE BEESBOOK PROJECT

In the BeesBook project, we aim to understand the complex social behavior of honey bees. For identification, a tag with a binary code is attached to the back of the bees.

The orientations in space, position, and bits of the tags are required to track the bees over time. Decoding this information is not trivial: the bees are oriented in any direction in space. The tag might be partially occluded. Moreover, spotlights on the tag can sometimes even confuse humans. A previously used computer vision pipeline did not perform reliably. Although we invested a substantial amount of time on labeling, a DCNN trained on this data did not perform significantly better (see Sec. 3). We therefore wanted to synthesize labeled images which are realistic enough to train an improved decoder network.

Following the idea outlined in section 3, we created a simple 3D model of a bee marker. The 3D model comprises a mesh which represents the structure of the marker and a simple camera model to project the mesh to the image plane. The model is parameterized by its position, its pitch, yaw and roll, and its ID. Given a parameter set, we obtain a marker image, a background segmentation mask

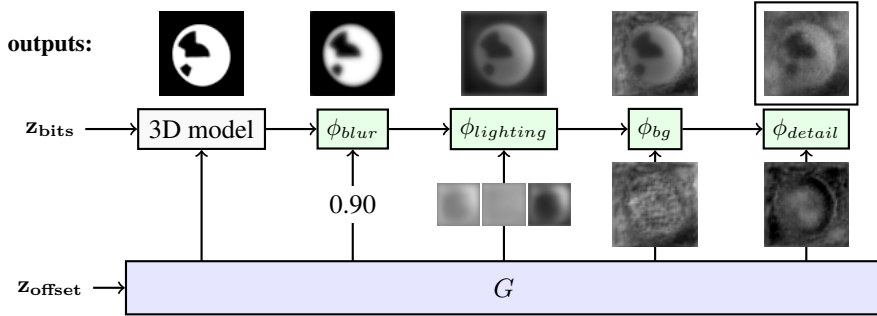

outputs:

Figure 5: Augmentation functions of the RenderGAN applied to the BeesBook project. The arrows from $G$ to the augmentation functions $\phi$ depict the inputs to the augmentation functions. The generator provides the position and orientations to the 3D model, whereas the bits are sampled uniformly. On top, the output of each stage is shown. The output of $\phi_{detail}$ is forwarded to the discriminator.

and a depth map. The generated images lack many important factors: blur, lighting, background, and image detail (see Fig. 2). A DCNN trained on this data does not generalize well (see Sec. 5).

Over the last years we collected a large amount of unlabeled image data. We successfully augmented the 3D model using this dataset, as described below.

We trained a neural network to emulate the 3D model. Its outputs are indistinguishable from the images of the 3D model. The discriminator error can now be backpropagated through the 3D model which allows the generator to also learn the distributions of positions and orientations of the bee marker. The IDs are sampled uniformly during training. The weights of the 3D model network are fixed during the RenderGAN training.

We apply different augmentation functions that account for blur, lighting, background, and image detail. The output of the 3D model and of each augmentation function is of shape $(64, 64)$ and in the range $[-1, 1]$. In Fig. 5, the structure of the generator is shown.

**Blurriness:** The 3D model produces hard edges, but the images of the real tags show a broad range of blur. The generator produces a scalar $\alpha \in [0, 1]$ per image that controls the blur.

$$\phi_{blur}(x, \alpha) = (1 - \alpha)(x - b_\sigma(x)) + b_\sigma(x) \tag{4}$$

where $b_\sigma(x) = x * k_\sigma$ denotes convolving the image $x$ with a Gaussian kernel $k_\sigma$ of scale $\sigma$. The implementation of the blur function is inspired by Laplacian pyramids (Burt & Adelson, 1983). As required for augmentation functions, the labels are preserved, because we limit the maximum amount of blur by picking $\sigma = 2$. $\phi_{blur}$ is also differentiable w.r.t the inputs $\alpha$ and $x$.

**Lighting of the tag:** The images from the 3D model are binary. In real images, tags exhibit different shades of gray. We model the lighting by a smooth scaling and shifting of the pixel intensities. The generator provides three outputs for the lighting: scaling of black parts $s_b$, scaling of white parts $s_w$ and a shift $t$. All outputs have the same dimensions as the image $x$. An important invariant is that the black bits of the tag must stay darker than the white bits. Otherwise, a bit could flip, and the label would change. By restricting the scaling $s_w$ and $s_b$ to be between 0.10 and 1, we ensure that this invariant holds. The lighting is locally corrolated and should cause smooth changes in the image. Hence, Gaussian blur $b(x)$ is applied to $s_b$, $s_w$, and $t$.

$$\phi_{lighting}(x, s_w, s_b, t) = x \cdot b(s_w) \cdot W(x) + x \cdot b(s_b) \cdot (1 - W(x)) + b(t) \tag{5}$$

The segmentation mask $W(x)$ is one for white parts and zero for the black part of the image. As the intensity of the input is distributed around -1 and 1, we can use thresholding to differentiate between black and white parts.

**Background**: The background augmentation can change the background pixels arbitrarily. A segmentation mask $B_x$ marks the background pixels of the image $x$ which are replaced by the pixels from the generated image $d$.

$$\phi_{bg}(x, d) = x \cdot (1 - B_x) + d \cdot B_x \tag{6}$$

The 3D model provides the segmentation mask. As $\phi_{bg}$ can only change background pixels, the labels remain unchanged.

**Details:** In this stage, the generator can add small details to the whole image including the tag. The output of the generator $d$ is passed through a high-pass filter to ensure that the added details are small enough not to flip a bit. Furthermore, $d$ is restricted to be in $[-2, 2]$ to make sure the generator cannot avoid the highpass filter by producing huge values. With the range $[-2, 2]$, the generator has the possibility to change black pixels to white, which is needed to model spotlights.

$$\phi_{detail}(x, d) = x + \text{highpass}(d) \tag{7}$$

The high-pass is implemented by taking the difference between the image and a blurred version of the image ($\sigma = 3.5$). As the spotlights on the tags are only a little smaller than the bits, we increase its slope after the cutoff frequency by repeating the high-pass filter three times.

The image augmentations are applied in the order as listed above: $\phi_{detail} \circ \phi_{background} \circ \phi_{lighting} \circ \phi_{blur}$. Please note that there exist parameters to the augmentation functions that could change the labels. As long as it is guaranteed that such augmentations will result in unrealistic looking images, the generator network will learn to avoid them. For example, even though the detail augmentation could be used to add high-frequency noise to obscure the tag, this artifact would be detected by the discriminator.

**Architecture of the generator:** The generator network has to produce outputs for each augmentation function. We will outline only the most important parts. See our code available online for all the details of the networks[1]. The generator starts with a small network consisting of dense layers, which predicts the parameters for the 3D model (position, orientations). The output of another dense layer is reshaped and used as starting block for a chain of convolution and upsampling layers. We found it advantageous to merge a depth map of the 3D model into the generator as especially the lighting depends on the orientation of the tag in space. The input to the blur augmentation is predicted by reducing an intermediate convolutional feature map to a single scalar. An additional network is branched off to predict the input to the lighting augmentation. For the background generation, the output of the lighting network is merged back into the main generator network together with the actual image from the 3D model.

For the discriminator architecture, we mostly rely on the architecture given by Radford et al. (2015), but doubled the number of convolutional layers and added a final dense layer. This change improved the quality of the generated images.

**Clip layer:** Some of the augmentation parameters have to be restricted to a range of values to ensure that the labels remain valid. The training did not converge when using functions like $\tanh$ or sigmoid due to vanishing gradients. We are using a combination of clipping and activity regularization to keep the output in a given interval $[a, b]$. If the input $x$ is out of bounds, it is clipped and a regularization loss $r$ depending on the distance between $x$ and the appropriate bound is added.

$$r(x) = \begin{cases} \gamma||x - a||_1 & \text{if } x < a \\ 0 & \text{if } a \leq x \leq b \\ \gamma||x - b||_1 & \text{if } x > b \end{cases} \tag{8}$$

$$f(x) = \min(\max(a, x), b) \tag{9}$$

With the scalar $\gamma$, the weight of the loss can be adapted. For us $\gamma = 15$ worked well. If $\gamma$ is chosen too small, the regularization loss might not be big enough to move the output of the previous layer towards the interval $[a, b]$. During training, we observe that the loss decreases to a small value but never vanishes.

**Training:** We train generator and discriminator as in the normal GAN setting. We use 2.4M unlabeled images of tags to train the RenderGAN. We use Adam (Kingma & Ba, 2014) as an optimizer with a starting learning rate of $0.0002$, which we reduce in epoch 200, 250, and 300 by a factor of $0.25$. In Fig. 6b the training loss of the GAN is shown. The GAN does not converge to the point where the discriminator can no longer separate generated from real samples. The augmentation functions might restrict the generator too much such that it cannot model certain properties. Nevertheless, it is hard for a human to distinguish the generated from real images. In some cases, the

---

[1]https://github.com/berleon/deepdecoder

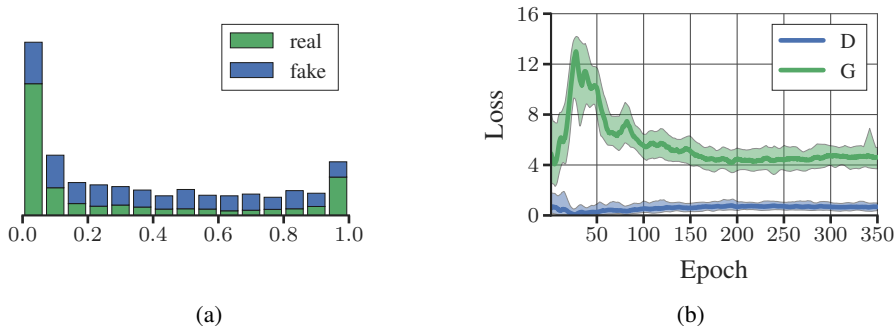

(a) (b)

Figure 6: **(a)** Histogram of the discriminator scores on fake and real samples. **(b)** Losses of the generator (G) and discriminator (D).

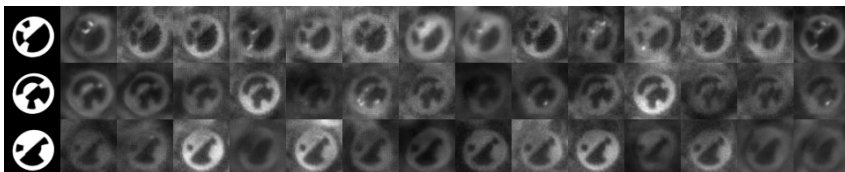

Figure 7: Random points in the z-space given the tag parameters

generator creates unrealistic high-frequencies artifacts. The discriminator unfailingly assigns a low score to theses images. We can therefore discard them for the training of the supervised algorithm. More generated images are shown in Appendix A. In Fig. 7, we show random points in the latent space, while fixing the tag parameters. The generator indeed learned to model the various lighting conditions, noise intensities, and backgrounds.

## 5 RESULTS

We constructed the RenderGAN to generate labeled data. But does a DCNN trained with the RenderGAN data perform better than one trained on the limited amounts of real data? And are learned augmentations indeed needed or do simple hand-designed augmentation achieve the same result? The following paragraphs describe the different datasets used in the evaluation. We focus on the performance of a DCNN on the generated data. Thus, we do not compare our method to conventional GANs as those do not provide labels and are generally hard to evaluate.

**Data from the *RenderGAN***: We generate 5 million tags with the RenderGAN framework. Due to the abundance, one training sample is only used twice during training. It is not further augmented.

***Real* Data**: The labels of the real data are extracted from ground truth data that was originally collected to evaluate bee trajectories. This ground truth data contains the path and id of each bee over multiple consecutive frames. Data from five different time spans was annotated – in total 66K tags. As the data is correlated in time (same ids, similar lighting conditions), we assign the data from one time span completely to either the train or test set. The data from three time spans forms the train set (40K). The test set (26K) contains data from the remaining two time spans. The ground truth data lacks the orientation of the tags, which is therefore omitted for this evaluation. Due to the smaller

Table 1: Datasets created with learned representations and hand-designed augmentations

| Name | Learned | Hand-Designed |
|---|---|---|
| HM 3D | 3D model | blur, lighting, background, noise, spotlights |
| HM LI | 3D model, blur, lighting | background, noise, spotlights |
| HM BG | 3D model, blur, lighting, background | noise, spotlights |

Table 2: Comparison of the mean Hamming distance (MHD) on the different data sets. More samples of the training data can be found in Appendix D.

| Data | MHD | Training Data |
|---|---|---|
| Real | 0.956 |  |
| HM 3D | 0.820 |  |
| HM LI | 0.491 |  |
| HM BG | 0.505 |  |
| RenderGAN | 0.424 |  |
| RenderGAN + Real | 0.416 |  |
| CV | 1.08 | |

size of the real training set, we augment it with random translation, rotation, shear transformation, histogram scaling, and noise (see Appendix C for exact parameters).

***RenderGAN + Real***: We also train a DCNN on generated and real data which is mixed at a 50:50 ratio.

**Handmade augmentations:** We tried to emulate the augmentations learned by the RenderGAN by hand. For example, we generate the background by an image pyramid where the pixel intensities are drawn randomly. We model all effects, i.e. blur, lighting, background, noise and spotlights (see Appendix B for details on their implementation). We apply the handmade augmentation to different learned representations of the RenderGAN, e.g. we use the learned lighting representation and add the remaining effects such as background and noise with handmade augmentations (*HM LI*). See Table 1 for the different combinations of learned representations and hand designed augmentations.

**Computer vision pipeline *CV* :** The previously used computer vision pipeline (Wario et al., 2015) is based on manual feature extraction. For example, a modified Hough transformation to find ellipses. The MHD obtained by this model is only a rough estimate given that the computer vision pipeline had to be evaluated and fine-tuned on the same data set due to label scarcity.

**Training setup**: An epoch consists of 1000 batches á 128 samples. We use early stopping to select the best parameters of the networks. For the training with generated data, we use the real training set as the validation set. When training on real data, the test set is also used for validation. We could alternatively reduce the real training set further and form an extra validation set, but this would harm the performance of the DCNN trained on the real data. We use the 34-layer ResNet architecture (He et al., 2015) but start with 16 feature maps instead of 64. The DCNNs are evaluated on the mean Hamming distance (MHD) i.e. the expected value of bits decoded wrong. Human experts can decode tags with a MHD of around 0.23.

**Results:** In Table 2, we present the results of the evaluation. The training losses of the networks are plotted in Fig. 8. The model trained with the data generated by the RenderGAN has an MHD of 0.424. The performance can furthermore be slightly improved by combining the generated with real data. The small gap in performance when adding real data is a further indicator of the quality of the generated samples.

If we use predictions from this DCNN instead of the computer vision pipeline, the accuracy of the tracking improves from 55% of the ids assigned correctly to 96%. At this quality, it is possible to analyze the social behavior of the honeybees reliably.

Compared to the handmade augmentations (*HM 3D*), data from the RenderGAN leads to considerably better performance. The large gap in performance between the HM 3D and HM LI data highlights the importance of the learned lighting augmentation.

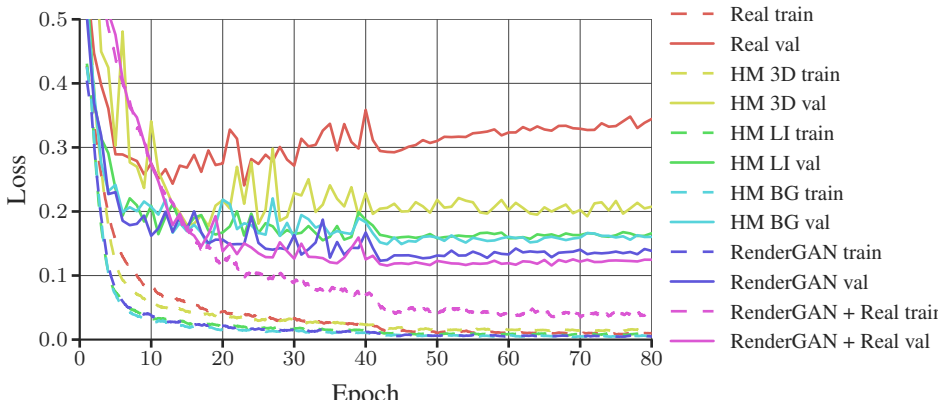

Figure 8: Training and validation losses of DCNNs trained on different data sets. As some data sets are missing the orientation of the tags, only the loss of the bits are plotted. Binary crossentropy is used as loss for the bits. The train and validation loss of each dataset have the same color.

## 6 DISCUSSION

We proposed a novel extension to the GAN framework that is capable of rendering samples from a basic 3D model more realistic. Compared to computer graphics pipelines, the RenderGAN can learn complex effects from unlabeled data that would be otherwise hard to model with explicit rules.

Contrary to conventional GANs, the generator provides explicit information about the synthesized images, which can be used as labels for a supervised algorithm. The training of the RenderGAN requires no labels.

We showed an application of the RenderGAN framework to the BeesBook project, in which the generator adds blur, lighting, background, and details to images from a basic 3D model. The generated data looks strikingly real and includes fine details such as spotlights, compression artifacts, and sensor noise.

In contrast to previous work that applied 3D models to produce training samples for DCNNs (Su et al., 2015; Richter et al., 2016; Ros et al., 2016), we were able to train a DCNN from scratch with only generated data that still generalizes to unseen real data.

While some work is required to adapt the RenderGAN to a specific domain, once set up, arbitrary amounts of labeled data can be acquired cheaply, even if the data distribution changes. For example, if the tag design changes to include more bits, small adaptions to the 3D model's source code and eventually the hyperparameters of the augmentation functions would be sufficient. However, if we had labeled the data by hand, then we would have to annotate data again.

While the proposed augmentations represent common image characteristics, a disadvantage of the RenderGAN framework is that these augmentation functions must be carefully customized for the application at hand to ensure that high-level information is preserved. Furthermore, a suitable 3D model must be available.

## 7 FUTURE WORK

For future work, it would be interesting to see the RenderGAN framework used on other tasks where basic 3D models exist e.g. human faces, pose estimation, or viewpoint prediction. In this context, one could come up with different augmentation functions e.g. colorization, affine transformations, or diffeomorphism. The RenderGAN could be especially valuable to domains where pre-trained models are not available or when the annotations are very complex. Another direction of future work might be to extend the RenderGAN framework to other fields. For example, in speech synthesis, one could use an existing software as a basic model and improve the realism of the output with a similar approach as in the RenderGAN framework.

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

# Appendices

## A  GENERATED IMAGES

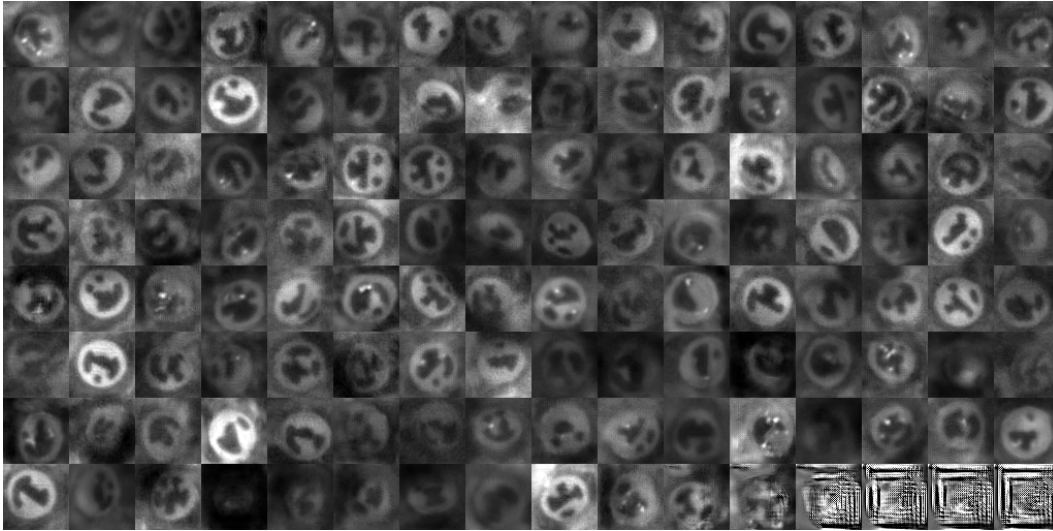

(a) Generated images

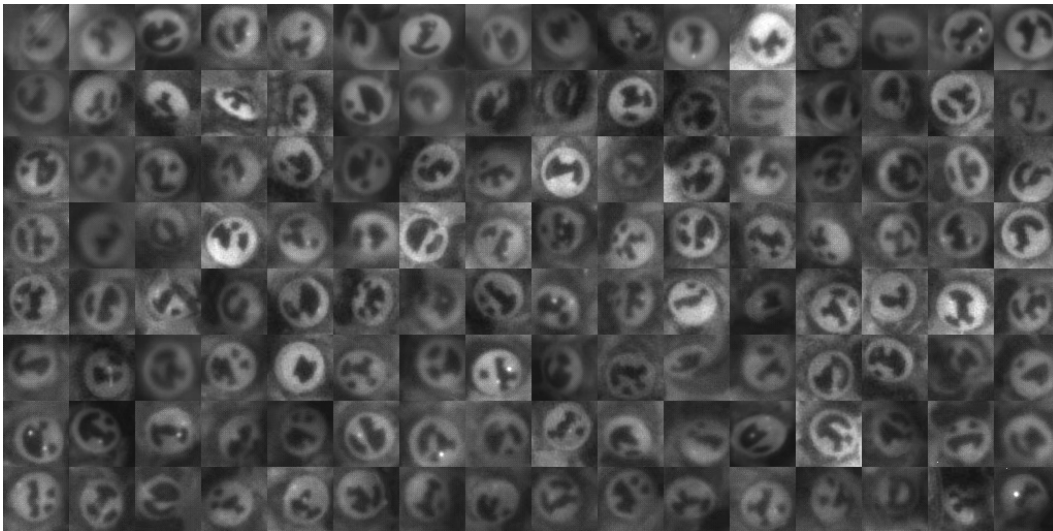

(b) Real images

Figure 9: Continuum visualization on the basis of the discriminator score: Most realistc scored samples top left corner to least realistc bottom right corner. Images with artifacts are scored unrealistic and are not used for training.

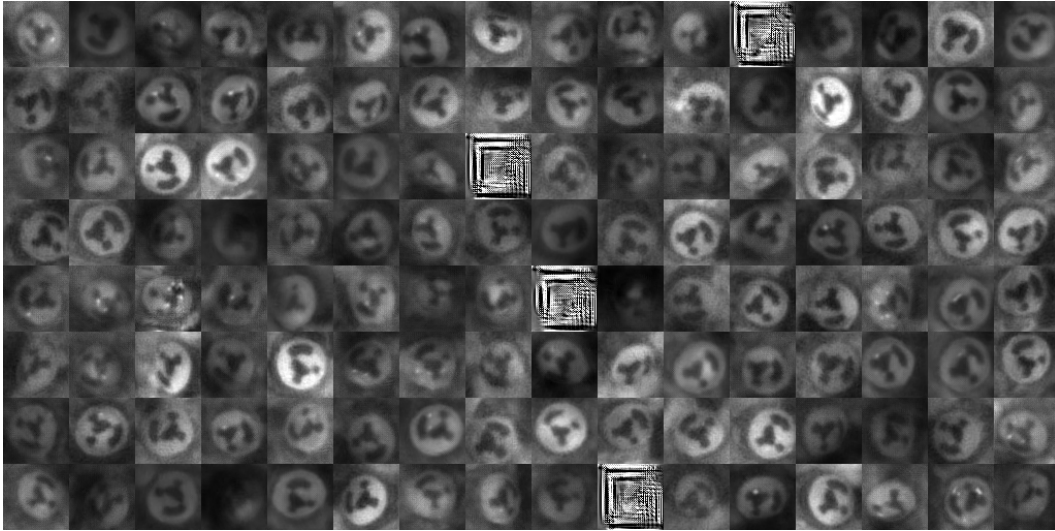

Figure 10: Images generated with the generator given a fixed bit configuration

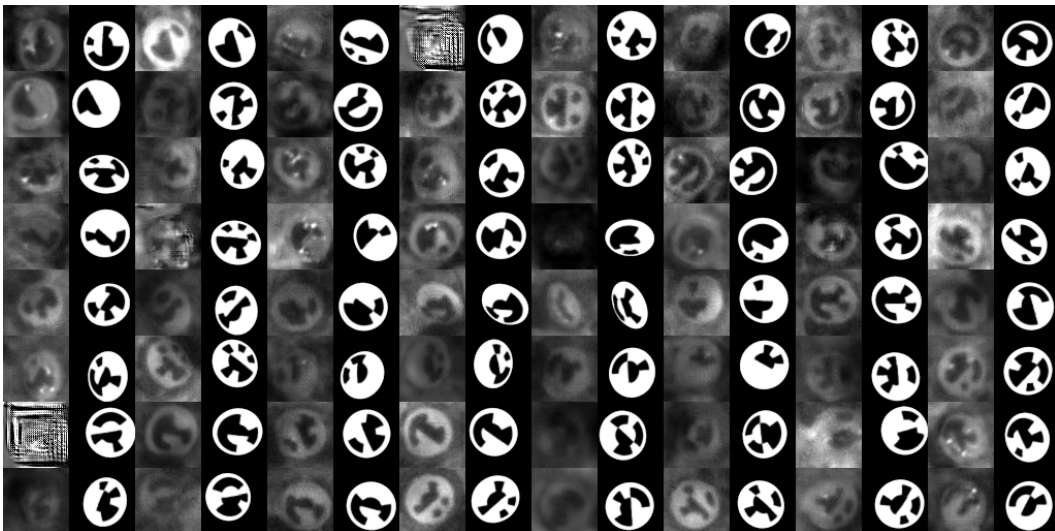

Figure 11: Correspondence of generated images and 3D model

## B  HANDMADE AUGMENTATIONS

We constructed augmentations for blur, lighting, background, noise and spotlights manually. For synthesizing lighting, background and noise, we use image pyramids, i.e. a set of images $L_0, \ldots, L_6$ of size $(2^i \times 2^i)$ for $0 \le i \le 6$. Each level $L_i$ in the pyramid is weighted by a scalar $\omega_i$. Each pixel of the different level $L_i$ is drawn from $\mathcal{N}(0, 1)$. The generated image $I_6$ is given by:

$$I_0 = \omega_0 L_0 \tag{10}$$

$$I_i = \omega_i L_i + \text{upscale}(I_{i-1}) \tag{11}$$

, where upscale doubles the image dimensions. The pyramid enables us to generate random images while controlling their frequency domain by weighting the pyramid levels appropriately.

- **Blur:** Gaussian blur with randomly sampled scale.
- **Lighting:** Similar as in the RenderGAN. Here, the scaling of the white and black parts and shifting is constructed with image pyramids.
- **Background:** image pyramids with the lower levels weight more.
- **Noise:** image pyramids with only the last two layer.
- **Spotlights**: overlay with possible multiple 2D Gauss function with a random position on the tag and random covariance.

We selected all parameters manually by comparing the generated to real images. However, using slightly more unrealistic images resulted in better performance of the DCNN trained with the HM 3D data. The parameters of the handmade augmentations can be found online in our source code repository.

## C  AUGMENTATIONS OF THE REAL DATA

We scale and shift the pixel intensities randomly, i.e. $sI + t$, where $I$ is the image and $s, t$ are scalars. The noise is sampled for each pixel from $\mathcal{N}(0, \epsilon)$, where $\epsilon \sim \max(0, \mathcal{N}(\mu_n, \sigma_n))$ is drawn for each image separately. Different affine transformations (rotation, scale, translation, and shear) are used.

Table 3: Parameters of the augmentation of the real data

| Name | Distribution |
|---|---|
| Intensity Scale (s) | unif(0.9, 1.1) |
| Intensity Shift (t) | unif(-0.2, 0.2) |
| Noise Mean ($\mu_n$) | 0.04 |
| Noise Std ($\sigma_n$) | 0.03 |
| Rotation | unif(0, $2\pi$) |
| Scale | unif(0.7, 1.1) |
| Shear | unif(-0.3, 0.3) |
| Translation | unif(-4, 4) |

## D  TRAINING SAMPLES

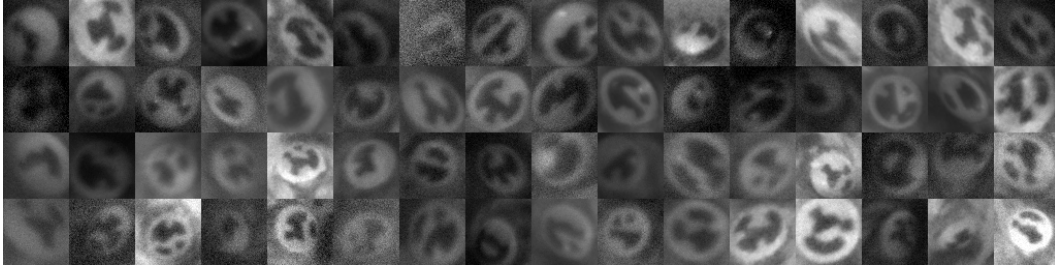

(a) Real trainings samples

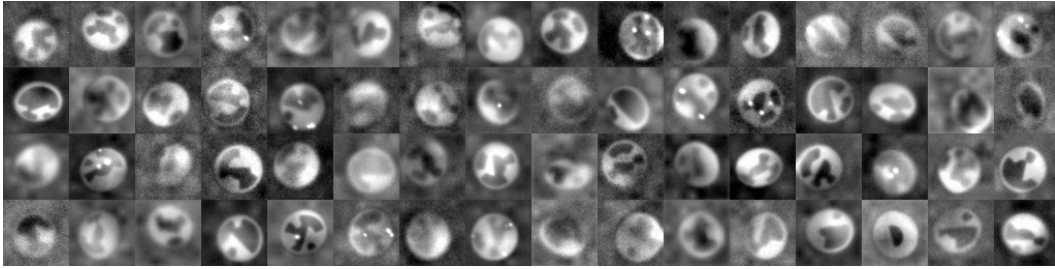

(b) HM 3D training samples

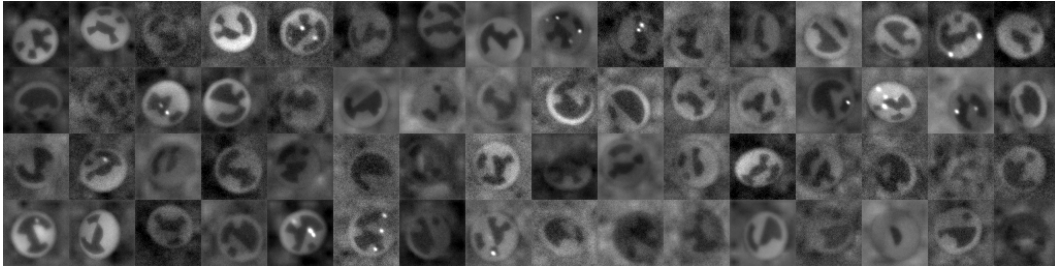

(c) HM LI training samples

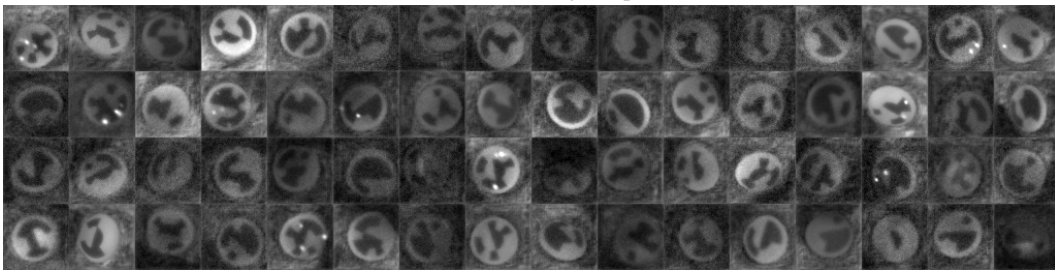

(d) HM BG training samples

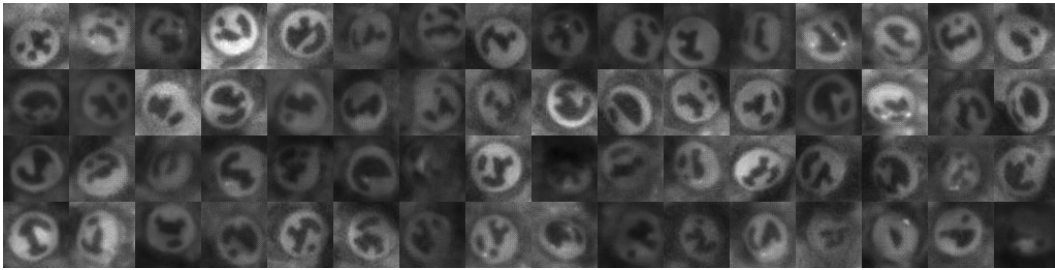

(e) RenderGAN

Figure 12: Training samples from the different datasets.

