# Peer review of "RenderGAN: Generating Realistic Labeled Data"

_ICLR 2017 — rejected_

[Author Response · Leon Sixt · 09 Dec 2016]
**Update paper**

We updated our paper based on the feedback from the pre-review questions.  We
included handmade augmentation in the evaluation.  We also retrained the DCNN on
the real data. Thanks for the feedback.

[Official Review · AnonReviewer1 · rating 6 · confidence 4 · 16 Dec 2016]

The paper proposes an approach to generating synthetic training data for deep networks, based on rendering 3D models and learning additional transformations with adversarial training. The approach is applied to generating barcode-like markers used for honeybee identification. The authors demonstrate that a classifier trained on synthetic data generated with the proposed approach outperforms both training on (limited) real data and training on data with hand-designed augmentations. 

The topic of the paper — using machine learning (in particular, adversarial training) for generating realistic synthetic training data — is very interesting and important. The proposed method looks reasonable, and the paper is written well. The downside is that experiments are limited to a fairly simple and not-widely-known domain of honeybee marker classification. While I am sure this is an important task by itself, in order to demonstrate general applicability of the method and to allow comparison with existing techniques, experiments on some standard and/or realistic datasets would be very helpful. Overall, I recommend acceptance, but encourage the authors to perform experiments on more datasets.

I appreciate that the authors added a baseline with manually designed transformations. This strengthens the paper.

As Reviewer3 points out, it would be interesting to analyze if restricting GAN to a fixed set of transformations is necessary here, and which transformations are most important. Perhaps this would provide some guidelines for designing sets of transformations for more complicated scenarios.

The authors should tone down their claims such as “Our method is an improvement over previous work  <...> Whereas previous work relied on real data for training using pre-trained models or mixing real and generated data, we were able to train a DCNN from scratch with generated data that performed well when tested on real data. “. This is not a fair comparison: the domain studied by authors in this work is much simpler than what was studied in these previous works, so this comparison is not appropriate.

[Official Review · AnonReviewer3 · rating 6 · confidence 4 · 16 Dec 2016]
**RenderGAN: Generating Realistic Labeled Data**

The submission proposes an interesting way to match synthetic data to real data in a GAN type architecture.
The main novelty are parametric modules that emulate different transformations and artefact that allow to match the natural appearance.

several points were raised during the discussion:

1. the proposed method is more model driven that previous GAN models. But does it pay off? how would a traditional GAN approach perform? The mentioned effects like blur, lighting and background could also potentially be modelled by upsamling network that directly predicts the image. I would assume that blur and lighting can be modelled by convolutions. transformations to some extend by convolutions - or spatial transformer networks.
The answers of the authors only partially addresses the point. The key proposal of the submission seems parameterised modules that can be trained to match the real data distribution. but it remains unclear why not a more generic parameterisation can also do the job. E.g. a neural network - as done in regular GANs. The benefit of introducing a stronger model is unclear. Using a render engine to generate the initial sample appearance if of limited novelty.


2. how does it compare to traditional data augmentation techniques, e.g. noise, dropout, transformations. you are linking to keras code - where data augmentation is readily available and could be tested (ImageDataGenerator)
The authors reply that plenty of such augmentation was used and more details are going to be provided in the appendix. it would have been appreciated if such information was directly included in the revision - so that the procedure could be directly checked. right now - this remains a point of uncertainty.

3. How do the different stages (\phis) effect performance? which are the most important ones?
The authors do evaluate the effect of hand tuning the transformation stages vs. learning them. it would be great to also include results of including/excluding stages completely - and also reporting how much the initial jittering of the data helps.

While there is an interesting idea of (limited) novelty to the paper, there are some concerns about evalations and comparisons as outlined above. In addition, only success on a single dataset/task is shown. Yet the task is interesting and seems challenging. Overall, this remains makes only a weak recommendation for acceptance.

[Official Review · AnonReviewer2 · rating 5 · confidence 3 · 17 Dec 2016]
**The proposed model has potential merits, but the paper is missing a critical baseline in the evaluation.**

This paper addresses the problem of decoding barcode-like markers depicted in an image.  The main insight is to train a CNN from generated data produced from a GAN.  The GAN is trained using unlabeled images, and leverages a "3D model" that undergoes learnt image transformations (e.g., blur, lighting, background).  The parameters for the image transformations are trained such that it confuses a GAN discriminator.  A CNN is trained using images generated from the GAN and compared with hand-crafted features and from training with real images.  The proposed method out-performs both baselines on decoding the barcode markers.

The proposed GAN architecture could potentially be interesting.  However, I won’t champion the paper as the evaluation could be improved.

A critical missing baseline is a comparison against a generic GAN.  Without this it’s hard to judge the benefit of the more structured GAN.  Also, it would be worth seeing the result when one combines generated and real images for the final task. 

A couple of references that are relevant to this work (for object detection using rendered views of 3D shapes):

[A] Xingchao Peng, Baochen Sun, Karim Ali, Kate Saenko, Learning Deep Object Detectors from 3D Models; ICCV, 2015.

[B] Deep Exemplar 2D-3D Detection by Adapting from Real to Rendered Views. Francisco Massa, Bryan C. Russell, Mathieu Aubry. CVPR 2016.

The problem domain (decoding barcode markers on bees) is limited.  It would be great to see this applied to another problem domain, e.g., object detection from 3D models as shown in paper reference [A], where direct comparison against prior work could be performed.  

I found the writing to be somewhat vague throughout.  For instance, on first reading of the introduction it is not clear what exactly is the contribution of the paper.  

Minor comments:

Fig 3 - Are these really renders from a 3D model?  The images look like 2D images, perhaps spatially warped via a homography.  

Page 3: "chapter" => "section".

In Table 2, what is the loss used for the DCNN?

Fig 9 (a) - The last four images look like they have strange artifacts. Can you explain these?

[Author Response · Leon Sixt · 05 Jan 2017]
**General Rebuttal**

Thank you very much for your reviews. Your feedback helped to improve the
manuscript significantly, and we are preparing a revised version of the
manuscript with changes outlined either below or in our responses to each
reviewer. Multiple valid points of criticism were raised during the review
process and have already been worked into the current version of the document.
For example, we included hand-designed augmentations for comparison with the
learned ones.

However, in two of the three reviews there seems to be a major misunderstanding
that we would like to clarify here. Since this relates to the central finding of
our paper, we would like to provide a detailed response to this point. We hope
that, in the light of this fact, the reviewer’s rating of our contribution’s
importance and novelty will be reconsidered.

> Reviewer 2: “A critical missing baseline is a comparison against a generic GAN.
> Without this it’s hard to judge the benefit of the more structured GAN.  Also,
> it would be worth seeing the result when one combines generated and real images
> for the final task.”

> Reviewer 3: “ [...] the proposed method is more model driven that previous GAN
> models. But does it pay off? how would a traditional GAN approach perform? [...]
> The answers of the authors only partially addresses the point. The key proposal
> of the submission seems parameterised modules that can be trained to match the
> real data distribution. but it remains unclear why not a more generic
> parameterisation can also do the job. E.g. a neural network - as done in regular
> GANs. The benefit of introducing a stronger model is unclear.”

The main point of critique here is that a comparison with a generic GAN
(Goodfellow et al. 2014) is missing. This comment implies that both methods (GAN
and RenderGAN) share the same task domain, which is incorrect. The task we
address is generating _labeled_ data. We emphasize that we do not refer to the
binary class label but rather to higher dimensional labels. In our example
scenario, this corresponds to images of bee markers and their respective bit
configuration (its ID) and rotations in 3D space. A generic GAN cannot generate
labels, it learns to generate realistic images _without_ labels. Ultimately, we
want to train a convnet (‘decoder network’) in a supervised setting to map an
image to its respective labels. Thus, we need labeled samples and hence,
a conventional GAN cannot be used as a baseline! Stated formally: the RenderGAN
samples from the joint distribution p(l, x) of labels l and data x whereas
a conventional GAN can only sample from the data distribution p(x).

There are two alternative approaches to our RenderGAN, one being a conventional
3-dimensional rendering pipeline that can be used to generate images of bee
markers with known ID and spatial orientation. Secondly, one could train the
decoder network with manually labeled data. Both approaches have been
implemented and tested against the RenderGAN and do not perform satisfyingly. To
improve both alternatives’ performance one would need to either tune the
rendering pipeline to match the details of the real world imaging process, or
label more data manually. Both measures are time-consuming and do not generalize
well when changing parts of the imaging process (lighting, cameras, compression,
etc.) or the marker design.

Our approach is to extend a generic GAN by adding several network modules, the
first being the network equivalent of a simple 3D model. Secondly, we learn
a number of parameterised augmentation functions. We would like to point out
that this approach was _not_ chosen to improve the generative capabilities of
the network but to constrain it in such a way that the image produced by the GAN
is correct with respect to the labels fed into the network. In our use case,
each image produced by the GAN has to preserve the given bit pattern and
rotation in space provided by the 3D model for the labels to remain valid. This
point was already addressed in our paper and the pre-review questions:

> Paper Introduction: “[...] We constrain the augmentation of the images such that
> the high-level information represented by the 3D model is preserved. The
> RenderGAN framework allows us to generate images of which the labels are known
> from the 3D model, and that also look strikingly real due to the GAN framework.
> The training procedure of the RenderGAN framework does not require any labels.
> We can generate high-quality, labeled data with a simple 3D model and a large
> amount of unlabeled data.”

> Our reply to Reviewer 3: “[...] The payoff is that we can generate labeled data
> with only a simple 3D model and unlabeled data. You are right. A DCGAN
> architecture can model all mentioned effects, even affine transformations. We
> trained a DCGAN on the data, and the quality of the synthesized images is
> similar. However, no labels can be collected in the conventional GAN framework.
> [...]”

All reviewers question the necessity of the constraints we introduced. One of
our early approaches was to add an offset to the 3D model, i.e. x = t + g(t)
where x is the synthesized image, t is an image from the 3D model, and g an
unconstrained generator. However, in our experiments, the generator learned to
synthesize realistic images but ignored the given template t completely. Thus,
no valid labels of the synthetic images could be collected. Since a decoder
network cannot be trained without labels, this approach cannot be used as
a baseline. We will revise our paper to clarify that an unconstrained GAN is not
a suitable baseline for our task.

[Author Response · Leon Sixt · 12 Jan 2017]
**Updated paper**

We uploaded a new version of the paper based on the peer review feedback.

* Clarified that an unconstrained GAN is not a suitable baseline.
* Added additional references pointed out by reviewer 2.
* Extracted a related work section to improve the overall clarity and also stated our contribution explicitly.
* Various modifications to improve clarity.

[Final Decision · Program Chairs · 06 Feb 2017]
**ICLR committee final decision**

This paper was fairly well received by the reviewers in terms of the underling idea but the fact that a very specialized problem was the focus of the paper held back reviewers from giving stronger ratings. The question of what sorts of baselines would be reasonable was discussed extensively as the reviewers felt that other credible baselines should be included. The authors argue certain baselines are not appropriate but they were not able to clearly sway the reviewers to a more positive rating based on their response to this issue. We recommend a workshop invitation for this paper.